# Associations between hyponatraemia, volume depletion and the risk of falls in US hospitalised patients: a case–control study

Elizabeth A Fehlberg,[1,2,3] Robert J Lucero,[1,2,4] Michael T Weaver,[1]
Anna M McDaniel,[1] A Michelle Chandler,[5] Phyllis A Richey,[6] Lorraine C Mion,[7]
Ronald I Shorr[2,8,9]

For numbered affiliations see end of article.

**Correspondence to**
Elizabeth A Fehlberg;
efehlberg@rti.org

## ABSTRACT

**Objective** We aimed to determine if abnormal laboratory values which may indicate volume depletion are associated with increased odds of experiencing a hospital-acquired fall.

**Design** Matched case–control study.

**Setting** Four hospitals located in the Southeast USA.

**Participants** Data from 699 adult fallers and 1189 matched controls (non-fallers) were collected via chart review from 2005 to 2010. Controls were matched to cases by nursing unit, time of fall and length of stay.

**Outcome measures** The primary exposures included serum sodium, blood urea nitrogen (BUN), creatinine, BUN/creatinine ratio and haematocrit. Conditional logistic regression with m:n matching was used to determine adjusted and unadjusted ORs.

**Results** Serum sodium levels were strongly associated with falls. In models controlling for demographic and other fall risk factors, patients with serum sodium levels of 125 mEq/L or less were associated with increased odds of experiencing a fall as compared with those with serum sodium levels of greater than 134 mEq/L (adjusted OR (aOR)=5.08, 95% CI 1.43 to 18.08). Conversely, elevated BUN, creatinine and elevated BUN/creatinine ratios were not associated with increased odds of experiencing a fall (aOR=0.64, 95% CI 0.49 to 0.84; aOR=0.70, 95% CI 0.54 to 0.92 and aOR=0.77, 95% CI 0.58 to 1.04, respectively.)

**Conclusions** Laboratory indices that may indicate volume depletion appear to be unrelated to falls. However, hyponatraemia does appear to be a risk factor for falls, and those with serum sodium levels below 126 mEq/L are at especially high risk. It may be that other deficits associated with hyponatraemia, like altered mental status, are associated with risk of experiencing a hospital-acquired fall. These results indicate that abnormal laboratory values, like low sodium, can be useful for identifying hospitalised patients at risk of falling. Therefore, further investigation into abnormal laboratory values as predictors of hospital-acquired falls is warranted.

## Strengths and limitations of this study

► The main strengths of this study include: (1) a large sample size collected over 6 years, (2) the setting which included four hospitals and (3) a matching strategy that controls for environmental and staffing factors at the time of a fall event.

► Only laboratory values that are not specific for volume depletion were used as potential indicators of volume depletion instead of other indicators such as urine output or orthostatic blood pressure.

► Our design did not account for all environmental factors like location of patient room relative to the nursing station.

► Due to the nature of observational research, we cannot exclude the possibility of incomplete control of potential confounders and there was a small number of patients with serum sodium levels of 125 mEq/L or less.

► The generalisability of our findings is limited by the context of our study and the sampling frame which relied on only patients whose laboratory results were known 24 hours prior to the fall index time.

In addition, falls are the leading cause of both fatal and non-fatal injuries among older adults.[1] Numerous studies have been conducted to examine the associations between patient-level risk factors and hospital-acquired falls. Examples of previously identified risk factors include dementia, diabetes, depression, cognitive status and impaired mobility.[3–9] Although many researchers have examined demographic factors and health conditions, few studies have aimed to understand the underlying biological mechanisms related to falling. Laboratory values, which have generally been neglected in previous falls research, can potentially increase our understanding of risk factors for hospital-acquired falls.[10]

Researchers have created tools to assist in the identification of patients at risk for falls

## INTRODUCTION

Each year, between 700 000 and 1 000 000 falls occur in US hospitals with associated total direct medical costs of $34 billion.[1 2]

including the Morse Fall Scale, the STRATIFY Scale and the Hendrich Fall Risk Model.[4 11 12] These tools include risk factors such as history of falls, mobility impairments and altered elimination. However, the predictability of these tools has been shown to vary considerably when subjected to external validations.[13] This indicates that there is a potential for improvement which could be achieved with the inclusion of additional factors like laboratory values. Researchers have examined the diagnosis of anaemia, haemoglobin levels and/or haematocrit levels as potential risk factors for hospital-acquired falls.[3 8 14–22] However, there has been limited examination of other laboratory values. Previous research has examined hyponatraemia in a psychiatric population, hyponatraemia and hypokalaemia in a Japanese cohort, albumin in a postoperative population, and creatinine, sodium and the ratio of blood urea nitrogen (BUN) to creatinine.[18 23–25] In addition, one study examined bivariate associations between approximately 30 laboratory values and falling.[17] However, all of the aforementioned non-anaemia laboratory work was limited in sample size, with approximately 100 fallers.

Previous research has identified volume depletion as a risk factor for falling, but to the best of our knowledge there has been limited examination of volume depletion and hospital-acquired falls. We found one study that examined the relationship between fluid intake and falls in a nursing home and found a significant decline in falls (p=0.05) during a hydration programme.[26] However, further examination of this potential relationship is warranted, especially since nursing home and hospital populations differ. Abnormal serum creatinine, BUN, sodium and haematocrit levels can indicate multiple health conditions, including volume depletion.[10 27–35] Specifically, abnormal laboratory values which may indicate volume depletion include high BUN levels (>21 mg/dL), high creatinine levels (>1.1 mg/dL in females and >1.2 mg/dL in males), high BUN to creatinine ratios (>20), high haematocrit levels (>45% in females and >51% in males) or high and low sodium levels (>145 mEq/L and <135 mEq/L, respectively).[36 37] To address whether volume depletion is potentially related to hospital-acquired falls, we used a case–control design to test if abnormal BUN, creatinine, haematocrit and sodium levels were associated with odds of a hospital-acquired fall.

## METHODS
### Conceptual framework
This study is informed by Choi's conceptual multisystemic fall prevention model.[38] Intrinsic risk factors, including physiological factors (ie, laboratory values), are directly associated with fall risk. In addition, symptoms secondary to physiological changes are associated with environmental and technological interventions. By discovering physiological intrinsic risk factors, modifications to the environment and care processes can theoretically lead to a reduction in falls.

### Study population
This case–control study was conducted using data that were collected between 2005 and 2010 across four hospitals. These hospitals are located in the Southeastern USA and belong to the same health system. Data from the largest hospital, a 635-bed urban community facility, were collected from 15 medical-surgical units. Data from the three other suburban community hospitals, ranging in size from 200 to 260 beds, were collected from nine medical-surgical units. A case was defined as an adult patient who fell on one of the 24 study units during the study period as reported in the hospital incident reporting system. In this study, a fall was defined as an unintentional change in position which resulted in coming to rest on the ground or a lower level. Falls resulting from catastrophic clinical events (eg, seizure, stroke or arrhythmia) were excluded. For each case, we identified up to two controls with a similar hospital length of stay that were on the same nursing unit at the same time the case fell. The data collection was approved by the Institutional Review Board of the University of Tennessee Health Science Center. This secondary analysis was approved by the University of Florida Institutional Review Board.

### Exposures
Exposures were determined for both cases and controls using medical records review by personnel blinded to the case status of the patient. We designated an index time for the controls based on the date and time that the matched case fell. This index time was used to establish the timeframe for collecting the exposures. Specifically, we recorded exposures most proximate to the time of the fall or the index time for controls. In addition, we excluded exposures that were greater than 24 hours before the fall event. Our primary exposures were serum sodium and other abnormal laboratory values that may indicate volume depletion including high BUN, creatinine, BUN to creatinine ratio and haematocrit. Volume depletion occurs when there is a decrease in circulating blood volume, which can occur as a result of inadequate fluid intake or excessive loss of fluids or blood. The abnormal laboratory value classifications can be found in table 2.[36]

### Covariates
Demographic covariates in this study included age, race and gender. We also adjusted for the medical conditions of Parkinson's disease, dementia, hypertension, congestive heart failure (CHF), diabetes mellitus and stroke. Presence of these comorbid conditions was defined as a positive history in the medical record. Also, we controlled for whether the patient had experienced an acute mental status change within the past 24 hours prior to the fall index time. In addition, we controlled for the patient's fall risk score prior to the fall index time. This score is based on the Morse Fall Scale with the inclusion of medications which potentially affect mobility or cognition (eg, sedatives, antipsychotics, antidepressants, diuretics and

opiates).[11] We based the binary measure of high fall risk (yes/no) on this standardised tool which is used across the four hospitals.

## Statistical analysis

Data analysis was performed using SAS software V.9.4 of the SAS System for Windows (SAS Institute, Cary, North Carolina, USA). Univariate descriptive statistics were calculated by fall status for the overall sample. Descriptive statistics including mean and SD were calculated for continuous data, and counts and percentages were calculated for categorical data. Potential multicollinearity issues were evaluated by testing the correlation between categorical exposures using the phi coefficient. Further, multicollinearity diagnostics were evaluated by checking condition index and variable inflation values. Since cases either had one or two controls, and were matched based on potential confounders, conditional logistic regression with m:n matching was used to determine bivariate and multivariable associations.[39 40] Using this approach, the unadjusted and adjusted ORs were estimated along with their corresponding 95% CIs. A series of multivariable models were created. Each multivariable model included one of the abnormal laboratory values of interest and the potential covariates. Potential covariates were included in the multivariable models regardless of bivariate p values.

Additional analyses were performed on serum sodium levels to further analyse the potential relationship between hyponatraemia and hospital-acquired falls. A categorical variable was created which included three levels of serum sodium including 125 mEq/L or lower, 126 to 134 mEq/L and 135 mEq/L or greater. These categories were created to determine if there was a dose–response association between hospital-acquired falls and sodium levels. Conditional logistic regression with m:n matching was used to determine the bivariate and multivariable associations. Using this approach, the unadjusted and adjusted ORs were estimated along with their corresponding 95% CIs. Two models were created including (1) an unadjusted model which only included the categorical sodium level variable and (2) an adjusted model which included the categorical sodium level variable and the covariates of high BUN, high creatinine, high BUN to creatinine ratio, age, race, gender, high fall risk score, acute mental status change, and the medical conditions of dementia, hypertension, CHF, diabetes and stroke.

## RESULTS

The final sample included 699 fallers and 1189 matched controls. Descriptive statistics by fall status for the final sample can be found in table 1. Controls were aged 62 years on average and 59.7% female, whereas the cases were aged 65 years on average and 50.9% female. In addition, only 35.8% of controls compared with 55.7% of cases were assessed as high risk prior to the fall index time. Also, 75.4% of cases had a diagnosis of hypertension, whereas 68.5% of controls had a diagnosis of hypertension. We did not include the exposures of Parkinson's disease, high haematocrit and high sodium in further analyses due to low frequencies of less than 3% in this sample. Specifically, 1.72% of cases and 1.01% of controls had a diagnosis of Parkinson's disease, 1.45% of cases and 1.09% of controls had high haematocrit levels, and 2.11% of cases and 2.40% of controls had high sodium levels.

Unadjusted ORs from bivariate conditional logistic regression models can be found in table 2. In the unadjusted model, the presence of low sodium increased the odds of a hospital-acquired fall (OR=1.485, 95% CI 1.136 to 1.940). In contrast, the presence of elevated BUN levels decreased the odds of a hospital-acquired fall (OR=0.785, 95% CI 0.622 to 0.991). The presence of elevated creatinine levels and BUN to creatinine ratios were not significantly associated with the occurrence of a hospital-acquired fall in the unadjusted models.

Adjusted ORs from the four multivariable conditional logistic regression models can also be found in table 2. Each of these models included one abnormal laboratory value which may indicate volume depletion while controlling for the covariates of age, race, gender, high fall risk score, acute mental status change, and the medical conditions of dementia, hypertension, CHF, diabetes and stroke. After adjusting for these covariates, the presence of low sodium significantly increased the odds of a hospital-acquired fall by 36% (p=0.04). Presence of high BUN and high creatinine were significantly associated with decreased odds of experiencing a hospital-acquired fall (OR=0.643, p=0.001, and OR=0.700, p=0.009, respectively). However, a high BUN to creatinine ratio was not significantly associated with experiencing a hospital-acquired fall in the adjusted model. The results of these four models indicate that based on abnormal laboratory values, volume depletion may not be associated with increased odds of a hospital-acquired fall.

Categories of serum sodium levels including 125 mEq/L or lower, 126 to 134 mEq/L and 135 mEq/L or greater were analysed further. Adjusted and unadjusted ORs from conditional logistic regression models for these categories of serum sodium levels can be found in table 3. Two models were created including an unadjusted model and a model adjusted for presence of high BUN, high creatinine, a high BUN to creatinine ratio, age, race, gender, high fall risk score, acute mental status change, and the medical conditions of dementia, hypertension, CHF, diabetes and stroke. In the unadjusted model, those with a serum sodium level of 125 mEq/L or lower were 4.82 times more likely to experience a hospital-acquired fall as compared with a serum sodium level of 135 mEq/L or greater (95% CI 1.54 to 15.12). In the adjusted model, those with a serum sodium level of 125 mEq/L or lower were 5.08 (95% CI 1.43 to 18.08) times more likely to experience a hospital-acquired fall. In addition, figure 1 displays the proportion of cases (fallers) and controls (non-fallers) among the three categories of serum sodium levels. Of those with a serum sodium level of 125 mEq/L or lower, approximately 71%

**Table 1** Characteristics of cases and controls by fall status (N=1888)

| Factor | n | Controls (n=1189) | | n | Cases (n=699) | | p Value* |
|---|---|---|---|---|---|---|---|
| | | Frequency (%) | Mean (SD) | | Frequency (%) | Mean (SD) | |
| Age | 1189 | | 61.6 (17.8) | 699 | | 64.9 (16.1) | <0.01 |
| Race | 1189 | | | 698 | | | <0.01 |
| White | | 538 (45.3) | | | 392 (56.2) | | |
| Not White | | 651 (54.8) | | | 306 (43.8) | | |
| Gender | 1189 | | | 698 | | | <0.01 |
| Male | | 479 (40.3) | | | 343 (49.1) | | |
| Female | | 710 (59.7) | | | 355 (50.9) | | |
| Hospital | 1187 | | | 699 | | | 0.979† |
| University | | 528 (44.5) | | | 320 (45.8) | | |
| Community 1 | | 285 (24.0) | | | 164 (23.5) | | |
| Community 2 | | 97 (8.2) | | | 57 (8.2) | | |
| Community 3 | | 277 (23.3) | | | 158 (22.6) | | |
| High fall risk | 1169 | 419 (35.8) | | 684 | 381 (55.7) | | <0.01 |
| Mental status change | 1183 | 224 (18.9) | | 692 | 180 (26.0) | | <0.01 |
| Medical conditions | | | | | | | |
| Parkinson's disease | 1185 | 12 (1.0) | | 698 | 12 (1.7) | | 0.14 |
| Dementia | 1185 | 139 (11.7) | | 698 | 106 (15.2) | | 0.06 |
| Hypertension | 1188 | 814 (68.5) | | 699 | 527 (75.4) | | <0.01 |
| CHF | 1186 | 252 (21.3) | | 694 | 160 (23.1) | | 0.35 |
| Diabetes | 1186 | 394 (33.2) | | 696 | 261 (37.5) | | 0.07 |
| Stroke | 1185 | 143 (12.1) | | 695 | 108 (15.5) | | 0.03 |

*p Values were determined using conditional logistic regression with m:n matching.
†Cases and controls were matched by unit and therefore matched by hospital as well.
CHF, congestive heart failure.

were fallers and 29% were non-fallers. In contrast, of those with a serum sodium level of 126 to 134 mEq/L and 135 mEq/L or greater, approximately 48% and 38% were fallers and 52% and 62% were non-fallers, respectively.

## DISCUSSION

In this study, we identified important associations between patient-level risk factors and hospital-acquired falls. Of the four variables chosen as indicators of volume depletion

**Table 2** Bivariate and multivariable conditional logistic regression of abnormal BUN, creatinine and sodium values on hospital-acquired falls

| Model | Factor | Controls (n) | Cases (n) | Unadjusted OR (95% CI) | Adjusted OR* (95% CI) |
|---|---|---|---|---|---|
| 1 | BUN>21 mg/dL | 337 | 213 | 0.785 (0.622 to 0.991) | 0.643 (0.494 to 0.838) |
| | BUN≤21 mg/dL | 579 | 411 | Reference | Reference |
| 2 | Cr>1.1 mg/dL† | 363 | 230 | 0.827 (0.655 to 1.045) | 0.700 (0.535 to 0.916) |
| | Cr≤1.1 mg/dL‡ | 556 | 397 | Reference | Reference |
| 3 | BUN/Cr>20 | 215 | 140 | 0.860 (0.657 to 1.124) | 0.766 (0.578 to 1.042) |
| | BUN/Cr≤20 | 700 | 483 | Reference | Reference |
| 4 | Na≥135 mEq/L | 762 | 469 | Reference | Reference |
| | Na<135 mEq/L | 155 | 148 | 1.485 (1.136 to 1.940) | 1.356 (1.013 to 1.816) |

*Conditional logistic regression model adjusted for age, race, gender, high fall risk score, acute mental status change, and medical conditions of dementia, hypertension, congestive heart failure, diabetes and stroke.
†In males>1.2 mg/dL.
‡In males≤1.2 mg/dL.
BUN, blood urea nitrogen; Cr, creatinine.

**Table 3** Bivariate and multivariable conditional logistic regression of serum sodium levels on hospital-acquired falls

| Serum sodium (mEq/L) | Controls (n) | Cases (n) | Unadjusted OR (95% CI) | OR adjusted for multiple risk factors* (95% CI) |
|---|---|---|---|---|
| 125 or lower | 5 | 12 | 4.82 (1.54 to 15.12) | 5.08 (1.43 to 18.08) |
| 126–134 | 150 | 136 | 1.37 (1.04 to 1.81) | 1.27 (0.94 to 1.73) |
| 135 or greater | 762 | 469 | Reference | Reference |

*Conditional logistic regression model adjusted for high BUN, high creatinine, high BUN to creatinine ratio, age, race, gender, high fall risk score, acute mental status change, and medical conditions of dementia, hypertension, congestive heart failure, diabetes and stroke.
BUN, blood urea nitrogen.

that were usable in the analyses, only low sodium was independently related to increased odds for falling in the hospital setting. However, the abnormal laboratory values of high BUN levels and high creatinine levels were significantly associated with decreased odds of experiencing a hospital-acquired fall. It is possible that high BUN and creatinine levels appear to be protective because high BUN and creatinine levels can result in fatigue.[36] A fatigued patient might ambulate less and therefore be less likely to experience a fall. Also, it should be noted that high BUN and creatinine levels can be indicative of conditions other than volume depletion, like a hypercatabolic state. In addition to potentially indicating volume depletion, electrolyte abnormalities like low sodium can occur secondary to other conditions like syndrome of inappropriate antidiuretic hormone secretion (SIADH). Hyponatraemia can result in deficits including altered cognitive status and weakness.[10 36] Lacking significant positive associations between other volume depletion-related laboratory values, it is likely that the relationship between hyponatraemia and hospital-acquired falls is related to deficits arising from hyponatraemia rather than directly related to volume depletion. In addition, we observed that the odds of experiencing a hospital-acquired fall appear to increase as sodium levels decrease independent

of high BUN, high creatinine, a high BUN to creatinine ratio and other risk factors. This pattern further suggests that hyponatraemia is a risk factor for experiencing a hospital-acquired fall and that volume depletion does not appear to be the casual pathway.

Our findings add to previous work examining the relationship between hyponatraemia and hospital-acquired falls. An observational study conducted in a single hospital in Japan with a sample containing 97 fallers found that when controlling for age, comorbidities and increases in sedative doses, hyponatraemia (serum sodium <135 mEq/L) significantly increased the odds of experiencing a hospital-acquired fall (OR=1.751).[25] In addition, an observational study examining hyponatraemia and hospital-acquired falls among a psychiatric population found that when controlling for age, antiepileptic drug use and selective serotonin reuptake inhibitor use, hyponatraemia significantly increased the odds of experiencing a hospital-acquired fall (OR=4.38).[24] In contrast, a case–control study found that low sodium was not significantly associated with experiencing a hospital-acquired fall in a population of only those aged 65 years and older.[23] However, this study was limited by sample size with only 62 fallers and 62 controls which may have limited the statistical power of its findings. Collectively, it is not easy to determine the relative contribution of our work because not all of these studies provided an operational definition of hyponatraemia.

The relationship between hyponatraemia and non-hospital acquired falls has also been examined in an observational study of fall-related and non-fall related geriatric trauma admissions. The investigators of that study determined that when controlling for potential confounders such as age and pre-existing conditions, patients with fall-related admissions were significantly more likely to have low sodium levels (OR=1.81).[41] Also, investigators using a case–control study design, including patients admitted to the emergency room with and without chronic hyponatraemia, found that patients with chronic hyponatraemia were significantly more likely to have experienced a fall after controlling for covariates (OR=67).[42] In a similar case–control study including geriatric patients that were admitted with and without hyponatraemia, investigators found that when controlling for covariates such as age, gender, admitting diagnosis and medications, patients with hyponatraemia were significantly more likely to have a fall associated with their admission (eg, as a presenting complaint) (OR=3.12).[43] In addition,

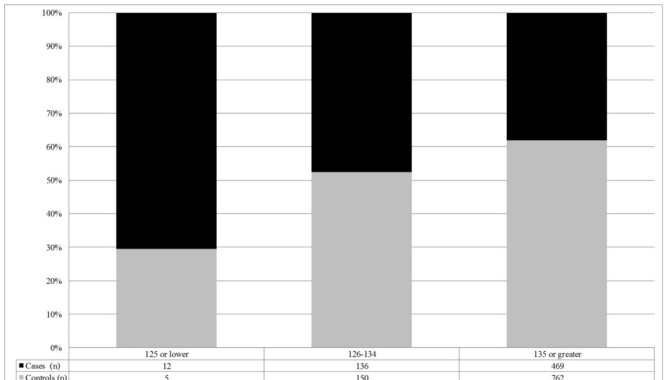

**Figure 1** Proportions of cases and controls among serum sodium levels. This figure displays three categories of serum sodium levels including 125 mEq/L or lower, 126 to 134 mEq/L and 135 mEq/L or greater. Of those cases and controls with these three different serum sodium levels, the black bars indicate the proportion that are cases and the grey bars indicate the proportion that are controls. Included at the bottom of the figure is the number of cases and controls within each category of serum sodium level.

investigators examining hyponatraemia among community-dwelling older adults found that after adjusting for age, gender and diuretic use, persons with hyponatraemia were significantly more likely to have experienced a fall (p=0.01).[44]

Unlike most previous work related to laboratory values and falls, our study is informed by a conceptual framework that hypothesised the relationship between laboratory values and fall risk.[38] Additional strengths of this study include the matching strategy and large sample size. Matching was based on nursing unit, date and time of fall, and length of stay which should control for environmental and staffing factors at the time of the fall. In previous research, measures of the hospital environment such as physical measures (eg, unit layout), resource availability measures (eg, nurse staffing) and culture measures (eg, magnet or teaching status) have been significantly associated with patient outcomes.[45–47] In addition, at the unit environment level, researchers have found that some medical units tend to have persistently low fall rates, whereas other medical units have persistently high fall rates.[47] The environment appears to have an effect on patient falls and it is, therefore, important to account for this variation. However, it should also be noted that our design cannot control for all environmental factors like location of patient room relative to the nursing station.

A limitation of this study is that only laboratory values were used as potential indicators of volume depletion. Further, the abnormal laboratory values of low sodium, high BUN, high creatinine and a high BUN to creatinine ratio are not specific to volume depletion but can also potentially indicate conditions other than volume depletion. For instance, hyponatraemia can occur as a result of SIADH. In addition, elevated BUN, creatinine and BUN to creatinine ratio levels can be seen in conditions such as CHF, sepsis, gastrointestinal obstruction and internal bleeding.[35] Although these abnormal laboratory values are not specific to volume depletion, they do have clinical value in determining whether a patient is volume depleted. For instance, increases in blood urea levels have been significantly associated with decreased hydration status.[34] In addition, if volume depletion is caused by vomiting, diarrhoea, diuretics or adrenal insufficiency, then the patient is likely to also be hyponatraemic.[29] No laboratory value gold standard exists for identifying volume depletion; however, the laboratory values used in this study are commonly used in clinical practice and have been used in prior research as markers for volume depletion.[30–33 35] Future work to further examine the potential relationship between volume depletion and hospital-acquired falls should consider using other potential indicators such as urine output, urine-specific gravity or orthostatic blood pressure. However, laboratory values are frequently collected in the hospital setting and are relatively reliable measures that have been neglected in previous falls research.

Additional limitations include that this was a secondary data analysis. This limited us to the exposure data available in the existing data set. Also, this study is limited by selecting for only those hospitalised patients who received laboratory results within their last 24 hours of hospitalisation prior to the fall index time. This, as well as sampling from one area of the country, limits the generalisability of these findings. However, the sample for this study was collected from four hospitals which contributes to the generalisability of these findings. Additional limitations include that observational research is susceptible to threats to internal validity including the incomplete control of potential confounders. However, we believe that our matching strategy helped to control for the relevant potential confounders of length of stay, nurse staffing, unit culture and unit environment.

## CONCLUSION

In this matched case–control study, we found a strong relationship between hyponatraemia and fall risk in hospitalised patients. This relationship is independent of increased BUN, creatinine and BUN to creatinine ratio as well as independent of demographic risk factors and other patient-level risk factors for hospital falls. Conversely, we found no other associations of laboratory findings consistent with volume depletion and increased risk of falls. It is possible that other indicators of volume depletion, like orthostatic hypotension, are necessary for the examination of this relationship. The results of this study do indicate that abnormal laboratory values, like low sodium, can be useful for identifying hospitalised patients at increased risk of experiencing a fall. Symptoms associated with hyponatraemia, including mental status changes, can be addressed with system-level and patient-level interventions like modifying the patient environment and regular patient surveillance. Further investigation into abnormal laboratory values as predictors of hospitalised-acquired falls is warranted, and if validated, should be added to currently used fall risk scales.

**Author affiliations**
[1]Departments of Biobehavioral Nursing and Family, Community, and Health System Science, University of Florida College of Nursing, Gainesville, Florida, USA
[2]Clinical and Translational Science Institute, University of Florida, Gainesville, Florida, USA
[3]Division of Research on Healthcare Value, Equity, and the Lifespan, RTI International, Research Triangle Park, NC, USA
[4]Center for Innovation on Disability and Rehabilitation Research (CINDRR), Malcom Randall VAMC, Gainesville, Florida, USA
[5]Methodist Healthcare University Hospital, Memphis, Tennessee, USA
[6]Department of Preventive Medicine, University of Tennessee Health Science Center, Memphis, Tennessee, USA
[7]Center of Excellence in Critical and Complex Care, The Ohio State University College of Nursing, Columbus, Ohio, USA
[8]Geriatric Research Education and Clinical Centers (GRECC), Malcom Randall VAMC, Gainesville, Florida, USA
[9]Department of Epidemiology, University of Florida, Gainesville, Florida, USA

**Acknowledgements**  Publication of this article was funded in part by the University of Florida Open Access Publishing Fund.

**Contributors**  All authors were involved in formulating the study concept/design. AMC, PAR and RIS were involved in the acquisition of the data and EAF, MTW and RIS performed the statistical analysis. EAF, RJL, MTW, AMMD, LCM and RIS were involved in the interpretation of the data, and all of the authors participated in the preparation of this manuscript. In addition, all authors approved the submitted manuscript.

**Funding** This work was supported by the National Institute on Aging (R01-AG033005 to RIS) and the National Center for Advancing Translational Sciences of the National Institutes of Health under University of Florida Clinical and Translational Science Awards (TL1TR001428 andUL1TR001427 to EAF).

**Disclaimer** The content is solely the responsibility of the authors and does not necessarily represents the official views of the National Institutes of Health.

**Ethics approval** Data collection was approved by the Institutional Review Board of the University of Tennessee Health Science Center. The secondary analysis was approved by the University of Florida Institutional Review Board.

**Provenance and peer review** Not commissioned; externally peer reviewed.

**Data sharing statement** No additional data are available for sharing.

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
