## [Reviewer comments · BMJ Open]

ARTICLE DETAILS

TITLE (PROVISIONAL)	Associations Between Hyponatremia, Volume Depletion, and the Risk of Falls in US Hospitalized Patients: A Case-Control Study
AUTHORS	Fehlberg, Elizabeth; Lucero, Robert; Weaver, Michael; McDaniel, Anna; Chandler, A.; Richey, Phyllis; Mion, Lorraine; Shorr, Ronald

VERSION 1 - REVIEW

REVIEWER	Anirban Ganguli Georgetown University-Washington Hospital Center Washington, DC USA
REVIEW RETURNED	05-Apr-2017

GENERAL COMMENTS	The authors present a well designed case control study to assess the utility of lab tests suggesting hypovolemia to predict the odds of experiencing a fall in hospitalized patients. While the study has a robust statistical analysis based on good methodological design a couple of assumptions are erroneous to allow appropriate clinical interpretation of numerical data. 1. Hyponatremia may or may not be secondary to hypovolemia. This is especially so in elderly hyponatremics where the incidence of euvolemic hyponatremia such as SIADH or reset osmostat are high2. Test characteristics of BUN, Cr or BUN/Cr ratio to define hypovolemia must be given to appropriately interpret results. The indicated references in the manuscript #10, 26-29 are not primary data but clinical reviews not clearly mentioning this assumption3. CKD itself is a major risk for falls or fractures- adjusted analysis does not mention this4. CHF, internal bleeding or hypercatabolic states can also elevate BUN/Cr ratio so this must be mentioned as caveats to using biochemical parameters to indicate hypovolemia5. At the best, based on data presented, it can be said that prevalent hyponatremia and not renal dysfunction increases the odds of falls in hospitalized patients when adjusting for well known risk factors.
--

REVIEWER	Mahesan Anpalahan Consultant Physician, Eastern Health. Clinical Associate Professor, Department of Medicine, The University of Melbourne and Monash University, Melbourne, Australia
REVIEW RETURNED	18-Apr-2017

GENERAL COMMENTS	Comments to authors: 1). Introduction and objective: a) The authors imply (page 6, lines 15-35) that no thorough
--

examination of volume depletion and hospital acquired fall was undertaken in previous studies and therefore they have designed this study to investigate this association more thoroughly. If this was their objective I'm afraid that the surrogate laboratory markers of fluid loss used in the study by themselves may not help to achieve this objective and the design of this study cannot be considered a significant improvement on previous studies. Although the laboratory markers of potential fluid loss used in this study, viz. creatinine, BUN, BUN to creatinine ratio and hyponatraemia may indicate fluid loss in the context of an appropriate clinical history, they often can be non-specifically abnormal in medical patients with multiple comorbidities. For example, hyponatraemia in medical patients often has been shown to have a SIADH aetiology, and this is not due to fluid loss. Similarly, medications and comorbidities such as heart failure and chronic kidney and liver diseases can lead to abnormal serum creatinine, BUN, BUN to creatinine ratio and hyponatraemia in the absence of fluid loss. This is an important weakness of this study and should be adequately discussed and emphasised. The authors may wish to include haematocrit as one of the laboratory variables along with the other four they have chosen, although haematocrit too can be affected by factors other than fluid loss. Surprisingly the study doesn't comment on patients with hypernatraemia – the authors may wish to incorporate this into the results and discussion. In fact, it is worth noting that hypernatraemia is more specific for fluid loss than hyponatraemia.

b) It may be useful to state how fall was defined in the study (were patients who suffered syncope, seizures and falls in the context of medication overdose or alcohol intoxication included?).

2) Study population:

In the context of the selection of controls, the authors state that they designated an index time for the controls based on the date and time that matched the case fell. It is unclear what the authors mean by this. Perhaps the authors may wish to explain this.

Also is there an age based inclusion criterion for the study?

3) Covariates:

a) It would be useful to state how some of the comorbidities were defined. For example, in the case of dementia, whether patients were formally screened for dementia during the admission, or was it defined on the basis of a positive history. Was delirium considered as part of dementia? It is not uncommon for older people, who do not have a history of dementia, to have delirium or acute confusion during hospital admission and whether this was factored into the assessment of comorbidities. Why admission diagnosis was not considered as one of the covariates?

4) Statistical analysis:

a) In the analyses that considered serum Na as a categorical variable with a cut off at 125mmols/l or lower, the authors have used two multivariable models- one adjusting for high BUN/ creatinine ratio and the other adjusting for various clinical variables. I wondered why all these covariates could not have been controlled for in the same model? Furthermore, of the three laboratory variables, the authors have adjusted only BUN/Cr in the multivariable model and have not included either serum Cr or BUN. The rationale for including only BUN/Cr in the model is unclear.

	b) The authors may wish to consider analysing the medical and surgical patients separately in order to minimise the confounding effects of unmeasured variables such as patient characteristics, admission diagnosis, in-hospital treatment etc., especially as controls were not matched for these variables. Furthermore, surgical patients are more likely to have received blood transfusions and intravenous fluid therapy and therefore one would expect the likely aetiology of hyponatraemia to vary significantly between these two groups of patients. 5) Results: a) The authors may consider providing the P values for the variables in table 1. b) Was the prevalence of fall by the severity of hyponatraemia statistically significant? (Figure 1, and Page 13, lines 3-14) 6) Discussion: The authors state (page 16, lines 15-20) that controls were matched for patient level risk factors. I'm not sure whether this is correct as they did not assess many patient level risk factors such as premorbid functional and mobility status, lower limb disabilities, history of previous falls, vision impairment, polypharmacy etc., although I agree that the controls were matched for environmental factors.
--	--

VERSION 1 – AUTHOR RESPONSE

RESPONSES TO REVIEWER 1: Anirban Ganguli

1. We completely agree that hyponatremia is not specific to hypovolemia, but can result from other conditions such as SIADH. We have added text throughout the manuscript in an attempt to clarify this limitation.

-STRENGTHS AND LIMITATIONS-section after abstract:

“Only laboratory values that are not specific for volume depletion were used as potential indicators of volume depletion...”

-INTRODUCTION-paragraph 3:

“Abnormal serum creatinine, BUN, sodium, and hematocrit levels can indicate multiple health conditions, including volume depletion”

-DISCUSSION-paragraph 1:

“In addition to potentially indicating volume depletion, electrolyte abnormalities such as low sodium can occur secondary to other conditions such as syndrome of inappropriate antidiuretic hormone secretion (SIADH).”

-DISCUSSION-paragraph 5:

“Further, the abnormal laboratory values of low sodium, high BUN, high creatinine, and a high BUN to creatinine ratio are not specific to volume depletion, but can also potentially indicate conditions other

than volume depletion. For instance, hyponatremia can occur as a result of syndrome of inappropriate antidiuretic hormone secretion (SIADH)”

2. We have added additional references where abnormal laboratory values have been used as markers for volume depletion. Additionally, we have added text describing these measures in the introduction and discussion, including their limitations.

-INTRODUCTION-paragraph 3:

“Abnormal serum creatinine, BUN, sodium, and hematocrit levels can indicate multiple health conditions, including volume depletion[10, 27-35]. Specifically, abnormal laboratory values which may indicate volume depletion include high BUN levels (>21mg/dL), high creatinine levels (>1.1mg/dL in females, >1.2mg/dL in males), high BUN to creatinine ratios (>20), high hematocrit levels (>45% in females and >51% in males), or high and low sodium levels (>145mEq/L and <135mEq/L, respectively)[36, 37].”

-DISCUSSION-paragraph 5:

“Although these abnormal laboratory values are not specific to volume depletion, they do have clinical value in determining whether a patient is volume depleted. For instance, increases in blood urea levels have been significantly associated with decreased hydration status[34]. Additionally, if volume depletion is caused vomiting, diarrhea, diuretics, or adrenal insufficiency, then the patient is likely to also be hyponatremic[29]. No laboratory value gold standard exists for identifying volume depletion, however the laboratory values used in this study are commonly used in clinical practice and have been used in prior research as markers for volume depletion[30-33, 35].”

3. Unfortunately, due to this being a secondary data analysis, we were unable to separate out CKD. We have added text highlighting the limitations of secondary data analysis.

-DISCUSSION-paragraph 6:

“Additional limitations include that this was a secondary data analysis. This limited us to the exposure data available in the existing dataset.”

4. We completely agree that we need to make this limitation clearer. Similar to hyponatremia, the laboratory values of BUN, Cr, and BUN to Cr ratio are also not specific to volume depletion. We have added text throughout the manuscript in an attempt to clarify these limitations. Additionally, we have added a paragraph to the end of the discussion further highlighting these limitations. We hope this will further increase the reader’s understanding of the limitations of our work.

-STRENGTHS AND LIMITATIONS-section after abstract:

“Only laboratory values that are not specific for volume depletion were used as potential indicators of volume depletion...”

-INTRODUCTION-paragraph 3:

“Abnormal serum creatinine, BUN, sodium, and hematocrit levels can indicate multiple health conditions, including volume depletion”

-DISCUSSION-paragraph 1:

“Also, it should be noted that high BUN and creatinine levels can be indicative of conditions other than volume depletion, such as a hypercatabolic state.”

-DISCUSSION-paragraph 5:

“A limitation of this study is that only laboratory values were used as potential indicators of volume depletion. Further, the abnormal laboratory values of low sodium, high BUN, high creatinine, and a high BUN to creatinine ratio are not specific to volume depletion, but can also potentially indicate conditions other than volume depletion. For instance, hyponatremia can occur as a result of syndrome of inappropriate antidiuretic hormone secretion (SIADH). Additionally, and elevated BUN, creatinine, and BUN to creatinine ratio levels can be seen in conditions such as congestive heart failure, sepsis, gastrointestinal obstruction, and internal bleeding[35]. Although these abnormal laboratory values are not specific to volume depletion, they do have clinical value in determining whether a patient is volume depleted. For instance, increases in blood urea levels have been significantly associated with decreased hydration status[34]. Additionally, if volume depletion is caused vomiting, diarrhea, diuretics, or adrenal insufficiency, then the patient is likely to also be hyponatremic[29]. No laboratory value gold standard exists for identifying volume depletion, however the laboratory values used in this study are commonly used in clinical practice and have been used in prior research as markers for volume depletion[30-33, 35].”

5. We do not necessarily wish to speculate about renal dysfunction. Instead, we were merely attempting to explore whether laboratory values could be useful for identifying patients at increased risk of falling. As we mentioned in the introduction and conclusion, current methods for identifying patients at risk of falling have room for improvement. Since hyponatremia was associated with increased odds of falling while controlling for well known risk factors, laboratory values such as sodium levels should be further investigated and potentially added to fall risk assessment tools.

RESPONSES TO REVIEWER 2: Mahesan Anpalahan

1a) We rephrased the last paragraph of the introduction to make our objective clearer, as well as the limitations to our study approach. Additionally, we completely agree that the laboratory values of serum sodium, BUN, Cr, and BUN to Cr ratio are not specific to volume depletion and that this limitation needs to be emphasized. We have added text throughout the manuscript in an attempt to

clarify this weakness to increase the reader's understanding of the limitations of our work. Also, we have added a paragraph to the discussion section dedicated to highlighting the limitations of using laboratory values as markers for volume depletion.

-Also, adding hematocrit is an excellent suggestion. We added high hematocrit as a potential predictor for volume depletion. Unfortunately, in this sample, having a high hematocrit level was rare. We have added text to the manuscript describing hematocrit as a potential predictor, and also stating that we could not use this predictor in regression analyses due to the low frequency in this sample (1.2%).

-Additionally, we completely agree that hypernatremia should be a potential predictor. However, we tried to use hypernatremia as a potential predictor, but unfortunately the frequency was too low in this sample. A statement to this effect can be found in the first paragraph of the results section.

-INTRODUCTION-paragraph 3:

"Previous research has identified volume depletion as a risk factor for falling, but to the best of our knowledge there has been limited examination of volume depletion and hospital-acquired falls. We found one study that examined the relationship between fluid intake and falls in a nursing home and found a significant decline in falls ($P=0.05$) during a hydration program[26]. However, further examination of this potential relationship is warranted, especially since nursing home and hospital populations differ. Abnormal serum creatinine, BUN, sodium, and hematocrit levels can indicate multiple health conditions, including volume depletion[10, 27-35]. Specifically, abnormal laboratory values which may indicate volume depletion include high BUN levels ($>21\text{mg/dL}$), high creatinine levels ($>1.1\text{mg/dL}$ in females, $>1.2\text{mg/dL}$ in males), high BUN to creatinine ratios (>20), high hematocrit levels ($>45\%$ in females and $>51\%$ in males), or high and low sodium levels ($>145\text{mEq/L}$ and $<135\text{mEq/L}$, respectively)[36, 37]. To address whether volume depletion is potentially related to hospital-acquired falls, we used a case-control design to test if abnormal BUN, creatinine, hematocrit, and sodium levels were associated with odds of a hospital-acquired fall."

-STRENGTHS AND LIMITATIONS-section after abstract:

"Only laboratory values that are not specific for volume depletion were used as potential indicators of volume depletion..."

-DISCUSSION-paragraph 1:

"In addition to potentially indicating volume depletion, electrolyte abnormalities such as low sodium can occur secondary to other conditions such as syndrome of inappropriate antidiuretic hormone secretion (SIADH)."

"Also, it should be noted that high BUN and creatinine levels can be indicative of conditions other than volume depletion, such as a hypercatabolic state."

-DISCUSSION-paragraph 5:

"A limitation of this study is that only laboratory values were used as potential indicators of volume depletion. Further, the abnormal laboratory values of low sodium, high BUN, high creatinine, and a high BUN to creatinine ratio are not specific to volume depletion, but can also potentially indicate conditions other than volume depletion. For instance, hyponatremia can occur as a result of syndrome of inappropriate antidiuretic hormone secretion (SIADH). Additionally, elevated BUN, creatinine, and BUN to creatinine ratio levels can be seen in conditions such as congestive heart failure, sepsis, gastrointestinal obstruction, and internal bleeding[35]. Although these abnormal laboratory values are not specific to volume depletion, they do have clinical value in determining whether a patient is volume depleted. For instance, increases in blood urea levels have been significantly associated with decreased hydration status[34]. Additionally, if volume depletion is caused vomiting, diarrhea, diuretics, or adrenal insufficiency, then the patient is likely to also be hyponatremic[29]. No laboratory value gold standard exists for identifying volume depletion, however the laboratory values used in this

study are commonly used in clinical practice and have been used in prior research as markers for volume depletion[30-33, 35].”

-RESULTS-paragraph 1:

“We did not include the exposures of Parkinson’s Disease, high hematocrit, and high sodium in further analyses due to low frequencies of less than 3% in this sample.”

1b) We have added additional text to describe how a fall was defined in the study.

-METHODS-Study Population:

“In this study, a fall was defined as an unintentional change in position which resulted in coming to rest on the ground or a lower level. Falls resulting from catastrophic clinical events (e.g., seizure, stroke, or arrhythmia) were excluded.”

2) Thank you for bringing this ambiguity to our attention. We understand the confusion that was created by the location of the description of the index time. We moved the description to the Exposure section and added additional text so that it would be clearer.

-Also, thank you for bringing the missing age information to our attention. Only adults were included in this study and there was no upper limitation on age. We have added text to clarify these criteria.

-METHODS-Exposures

“We designated an index time for the controls based on the date and time that the matched case fell. This index time was used to establish the timeframe for collecting the exposures. Specifically, we recorded exposures most proximate to the time of the fall, or the index time for controls. Additionally, we excluded exposures that were greater than 24 hours before the fall event.”

-METHODS-Study Population

“A case was defined as an adult patient...”

3a) We have added text to clarify how a comorbidity was defined.

-Delirium is an excellent point. We added in a covariate which indicates whether the patient experienced an acute mental status change within the past 24 hours. However, we were concerned that this might create collinearity issues in the models. Therefore, we examined model collinearity diagnostics and determined that collinearity was not an issue.

- Unfortunately, data on admission diagnosis was not originally collected. We have added text highlighting the limitations of secondary data analysis.

-METHODS-Covariates

“Presence of these comorbid conditions was defined as a positive history in the medical record.”

“Also, we controlled for whether the patient had experienced an acute mental status change within the past twenty-four hours prior to the fall index time”

-METHODS-Statistical Analysis

“Further, multicollinearity diagnostics were evaluated by checking condition index and variable inflation values.”

-DISCUSSION-paragraph 6:

“Additional limitations include that this was a secondary data analysis. This limited us to the exposure data available in the existing dataset.”

4a) We changed the analysis so now there is only one multivariable model which adjusts for high BUN, high Cr, high BUN/Cr, and the various clinical variables. Once again, we were concerned that this might create collinearity issues in the models. Therefore, we examined model collinearity diagnostics and determined that collinearity was not an issue.

4b) Separately analyzing medical and surgical patients is a very interesting idea. Unfortunately, this is a secondary analysis and data on whether the patients were primarily medical or surgical was not originally collected.

5a) We have added P-values to Table 1

5b) The categorical serum sodium variable (125 or less, 126-134, and 135 or greater) was significant overall in unadjusted ($P=0.002$) and adjusted models ($P=0.0138$). The odds ratios and confidence intervals for the individual categories can be seen in Table 3. Additionally, decreasing sodium levels were also significant when we analyzed sodium as a continuous predictor (aOR=1.052; 95% CI=1.022-1.082). We have decided not to include sodium as a continuous predictor for simplicity in interpretation of our findings.

6) We have removed the following statement from the manuscript: “Thus, this matching strategy is best for identifying patient-level fall risk factors.”

VERSION 2 – REVIEW

REVIEWER	Anirban Ganguli Staff Nephrologist Georgetown University/Washington Hospital Center Washington, DC USA
REVIEW RETURNED	16-May-2017

GENERAL COMMENTS	The manuscript in its present form is much improved . I have only one minor comment to make. I did not raise the issue of hematocrit in my initial review since I was not sure if the authors could relook data to present this as another surrogate marker of hypovolemia. I agree that if the number of patients with raised hematocrit are inadequate it is improper to conduct regression analysis. However a crude test of proportions between cases and control for patients in whom this data is available could be very informative.
---

REVIEWER	Mahesan Anpalahan Consultant Physician, Eastern Health. Clinical Associate Professor, Department of Medicine, The University of Melbourne and Monash University, Melbourne, Australia
REVIEW RETURNED	21-May-2017

GENERAL COMMENTS	The authors have now addressed most of my concerns. However, I believe that they should be more circumspect about some of their conclusions. For example, they conclude that volume depletion appears to be unrelated to falls whereas hyponatraemia does appear to be a risk factor for falls. I'm not sure that the data presented allow us to make assertions about volume depletion. The conclusion may be amended as follows: “Laboratory indices that may indicate volume depletion appear to be unrelated to falls-----“.
--

VERSION 2 – AUTHOR RESPONSE

REVIEWER 1 (Anirban Ganguli) COMMENT: I did not raise the issue of hematocrit in my initial review since I was not sure if the authors could relook data to present this as another surrogate marker of hypovolemia. I agree that if the number of patients with raised hematocrit are inadequate it is improper to conduct regression analysis. However a crude test of proportions between cases and control for patients in whom this data is available could be very informative.

RESPONSE TO REVIEWER 1: Since we have an unevenly (m:n) matched sample, we used generalized linear mixed models (GLMM) with accommodation for matching and a logit link function to test whether there was a significant association between fall status and presence of high hematocrit. The odds for high hematocrit levels were similar between faller and non-faller groups (p<0.533). Nevertheless, we understand the desire to provide more information about high hematocrit. To this end, we have added additional text in the manuscript which includes the percentages of cases and controls with high hematocrit levels.

RESULTS-Paragraph 1

"...1.45% of cases and 1.09% of controls had high hematocrit levels..."

REVIEWER 2 (Mahesan Anpalahan) COMMENT: The authors have now addressed most of my concerns. However, I believe that they should be more circumspect about some of their conclusions. For example, they conclude that volume depletion appears to be unrelated to falls whereas hyponatraemia does appear to be a risk factor for falls. I'm not sure that the data presented allow us to make assertions about volume depletion. The conclusion may be amended as follows: "Laboratory indices that may indicate volume depletion appear to be unrelated to falls-----".

RESPONSE TO REVIEWER 2: Thank you for bringing this conclusion statement to our attention. We have corrected the conclusion statement as you suggested.

-ABSTRACT-Conclusions:

"Laboratory indices that may indicate volume depletion appear to be unrelated to falls. However, hyponatremia does appear..."

VERSION 3 – REVIEW

REVIEWER	Mahesan Anpalahan Consultant Physician, Eastern Health. Clinical Associate Professor, Department of Medicine, The University of Melbourne and Monash University, Melbourne, Australia
REVIEW RETURNED	12-Jun-2017

GENERAL COMMENTS	My comments/queries have been addressed
---